# Investigate Sex Dimorphism of Cerebral Myelination Across Lifespan by Leveraging Conditional Variational Autoencoder

**Jinghang Li**[1]                                                                          JIL202@PITT.EDU
**Linghai Wang**[1]                                                                       L.WANG@PITT.EDU
**Chang-le Chen**[1]                                                                     CHC348@PITT.EDU
**Tamer Ibrahim**[1,2,]                                                                TIBRAHIM@PITT.EDU
**Howard Aizenstein**[1,2]                                                          AIZEN@PITT.EDU
**Minjie Wu**[1,2]                                                                         MIW75@PITT.EDU

[1] *Department of Bioengineering, University of Pittsburgh, Pittsburgh, PA, USA*

[2] *Department of Psychiatry, University of Pittsburgh School of Medicine, Pittsburgh, PA, USA*

**Editors:** Under Review for MIDL 2023

## Abstract

In this work we investigated the potential sex differences in white matter aging using conditional variational autoencoder (cVAE) on myelin content MR images. The cVAE model was trained along with a supervised brain age prediction model, which learns the representation of myelination aging process within a single end-to-end model architecture. The training was conducted on a normal aging dataset (CamCAN) that included 708 individual MR images. Our brief exploration revealed that women might have slightly less white matter myelination than men do at an older age. Additionally, our brain age prediction model suggested different aging regressions for men and women.

**Keywords:** Conditional VAE, Brain age prediction, Sex differences, Myelin map

## 1. Introduction

Many efforts have been made in leveraging generative models such as variational autoencoders (VAE) in neuroimaging analysis (Zhao et al., 2019). These models are capable of learning the complex distributions of heterogeneous neuroimages in an unsupervised manner. In this work, we investigated the white matter aging differences between men and women by using conditional variational autoencoder (cVAE) on myelin content MR images called T1-w/T2-w ratio (Li et al., 2021). In contrast to the traditional VAE model, we incorporated a supervised brain age prediction model in the overall training process to constrain the conditional distribution in the latent space. This unique design can not only provide a brain age prediction model but also learn the representation of myelination aging process within a single end-to-end model architecture.

## 2. Method and Data

We used the publicly available CamCAN dataset that included 708 (359 female, 349 male) individual T1-w and T2-w MR structural scans acquired at 3 Tesla magnetic field strength. Both T1-w and T2-w MR images are of 1-mm isotropic resolution. T1-w/T2-w ratio

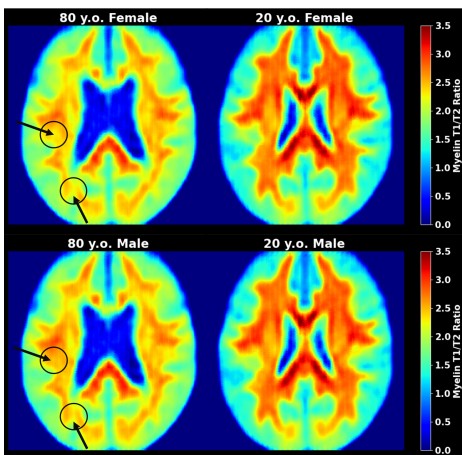

*Figure 1: Generated ratio maps from separately trained male and female cVAE models conditioned on age. We observed significant ventricle enlargement across sex as well as a distinctive decrease in myelin content with aging. Additionally, female might have slightly worse white matter health than male does at an older age. The regions with less myelin content were circled in the figure.*

images were acquired following the pipeline established in (Li et al., 2021). We then built the 3D cVAE model that encoded input images of size (160x160x160). In our generative model, $p(x^n, z^n, c^n)$, we have $X = \{x^{(1)}, x^{(2)}...x^{(n)}\}$, $C = \{c^{(1)}, c^{(2)}...c^{(n)}\}$, and $Z = \{z^{(1)}, z^{(2)}...z^{(n)}\}$, where X is the training images, C is the subjects' chronological age, and Z is the encoded latent space vectors. With the additional brain age predictor, we then have the following loss function:

$$L = -D_{KL}(q(z|x,c) \,||\, p(z|c)) + \underset{q(z|x,c)}{\mathbb{E}}[log\, p(x|z,c)] + \mathcal{L}_{MSE} \qquad (1)$$

where $\mathcal{L}_{MSE} = \Sigma_n(\hat{y}_n - y_n)^2$, with $y_n$ being the chronological age and $\hat{y}_n$ being the predicted age. Unlike vanilla VAE, in the cVAE model, the latent representation vector z was sampled from a conditioned distribution $p(z|c)$, where c is the one-hot encoded vector for age. To better disentangle the latent vector on both age and sex, we trained one model on male images only and one on female images only. By leveraging the reparameterization trick (Zhao et al., 2019), the cVAE model was trained with a learning rate of 0.0003 for 200 epochs. The model was implemented using PyTorch, and the training was carried out on NVIDIA A100 40GB at the University of Pittsburgh Center for Research Computing.

## 3. Results and Discussion

In this study, we explored the sex differences between men and women on white matter myelination maps using a normal aging dataset. Figure 1 shows the aging patterns as well as the sex differences in white matter integrity generated by the model after training. Distinctively, ventricle size increases drastically with age. Moreover, we discovered the decrease in myelin sheath as we age across sex; particularly, women might have slightly less white matter myelination than men do at an older age. Figure 2 shows the brain age

prediction result using the trained male model conducting inferences on both male and female ratio maps. The two fitted regression models suggest different aging pattern for men and women. Specifically, the results indicate that women ratio images appear to be older than that of men at an older age. Further region of interest (ROI) based analysis should be considered to reveal the specific underlying aging distinctions in terms of sex dimorphism. This study provided a brief investigation on white matter myelination across the lifespan and between sex. The results suggest that generative models such as conditional VAE can also serve as a normative model to quantify the spatial-temporal patterns of brain aging on myelination.

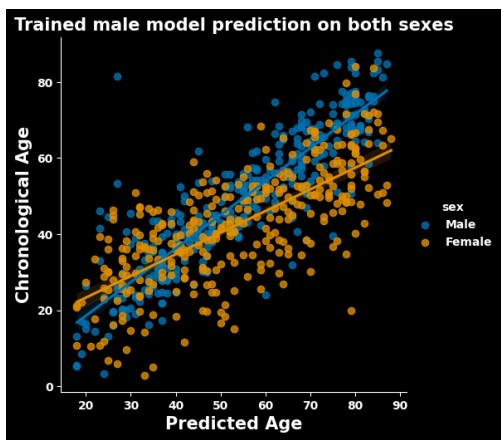

*Figure 2: Fitted linear regression models on predicted brain age using both male and female ratio images. The inference was done on the trained male age prediction model. The fitted regression models suggest that female myelin content appears to be older than that of male.*

## Acknowledgments

This research was supported in part by the University of Pittsburgh Center for Research Computing. Specifically, this work used the GPU cluster which is supported by NIH award number R01-AG063525

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
