# OpenReview forum: "Investigate Sex Dimorphism of Cerebral Myelination Across Lifespan by Leveraging Conditional Variational Autoencoder"
_MIDL.io/2023/Short_Paper_Track — MIDL 2023 Short paper track Poster_

### Official Review · Reviewer_xEvS · 2023-04-10
**Unclear method and results**

**Rating:** 6
**Confidence:** 4

**Review:**

The authors trained a conditional VAE (cVAE) to study myelin ratio images as a function of age, while also regressing age from the data. The paper is decently written and presents an interesting application of cVAEs to aging research. Issues with this submission include:
- Even if space is limited (but authors still had space for one more paragraphs since references were excluded from the page limit), some crucial details of the method should have been included, e.g., what variables age was regressed from (the images? The latents? both); or what the dimensionality of z was.
- The differences between males and females in figure 1 are qualitative, when it would have been quite easy to compare then quantitatively (are they even statistically significant?)
- Figure 2: do differences disappear if you train a regressor for females and another for males? Also: why are both regression lines below the identity y=x (both in terms of slope and intercept).

---

### Official Review · Reviewer_Pcq5 · 2023-04-24
**Interesting application but unclear methods/contribution attribution**

**Rating:** 5
**Confidence:** 4

**Review:**

This paper proposes using a conditional VAE on myelin content MR images trained with brain age prediction to learn representation of myelination aging process in single model. Training was conducted using a single aging dataset and used to assess sex differences in myelination and aging.

Strengths:
+ Interesting topic on myelination aging with a focus on sex differences
+ Uses considerable size public dataset (>700 subjects) for experiments
+ Incorporates brain age prediction into training to learn a representation for myelin images with consideration of age

Weaknesses:
- Method contribution unclear - likely application of Zhao et al (in refs) model to myelin images (instead of T1-weighted in Zhao et al.), which would be fine, but it is unclear and if so credit should be given
- Unclear what the model architecture is like other than is a "conditional VAE". E.g., what is encoder/decoder structure? What features/level of the model is used for the brain age prediction ?
- Unclear experimental settings, e.g., how is data split fo training / test? Notes train 2 separate models for males vs. females but is all data used for fitting or is cross validation performed? So are results from test data or train data? Important to understand generalizability of results/model.